# Trends and Sex Differences in Hospitalizations and Mortality in Parkinson’s Disease in Spain (2010–2019): A Nationwide Population-Based Study

**DOI:** 10.3390/jcm12030902

**Published:** 2023-01-23

**Authors:** Víctor Gómez-Mayordomo, Fernando Alonso-Frech, Valentín Hernández-Barrera, David Carabantes-Alarcon, José J. Zamorano-León, Ana Lopez-de-Andrés, Rodrigo Jiménez-García

**Affiliations:** 1Department of Neurology, Hospital Clínico San Carlos, San Carlos Research Health Institute (IdISSC), 28040 Madrid, Spain; 2Department of Neurology, Institute of Neurosciences, Hospital Universitario Vithas La Milagrosa, Vithas Hospital Group, 28010 Madrid, Spain; 3Preventive Medicine and Public Health Teaching and Research Unit, Health Sciences Faculty, Universidad Rey Juan Carlos, 28922 Alcorcón, Spain; 4Department of Public Health and Maternal & Child Health, San Carlos Research Health Institute (IdISSC), Faculty of Medicine, Universidad Complutense de Madrid, 28040 Madrid, Spain

**Keywords:** Parkinson´s disease, mortality, hospitalization, sex differences, epidemiology

## Abstract

The incidence of hospitalizations of Parkinson´s disease (PD) in Spain suffered a steady rise from 1997 to 2012. However, data on the trends during the following decade (2010–2019) are lacking. Hospital admissions with a primary and secondary diagnosis of PD were selected using the Spanish National Hospital Discharge Database (SNHDD) for the period 2010–2019. The primary endpoint was the incidence of hospitalizations and in-hospital mortality, stratified in biannual periods. The incidence of PD hospitalizations increased progressively over time from 81.25 cases in 2010–2011 to 94.82 cases in 2018–2019 per 100,000 inhabitants. Male sex, age and comorbidity also increased progressively in PD inpatients. PD as a comorbid condition presented a higher increment (annual percentage of change, APC +1.71%, *p* < 0.05) than PD as the main reason of hospitalization (APC +1.26%, *p* < 0.05). In the multivariate regression model, factors associated with mortality were male sex (OR = 1.15, 95% CI 1.01–1.35), age (>80 years, OR = 12.76, 95% CI 3.96–29.64) and comorbidity (Charlson index ≥ 2, OR 1.77, 95% CI 1.69–1.85). Adjusted mortality by age, sex, comorbidity and diagnostic position remained stable. In conclusion, PD hospitalizations in Spain have increased, with a parallel increment in mean age, male sex and higher comorbidities. However, adjusted mortality remains unchanged. The burden of this disease may increase the complexity and costs of hospital care in the future.

## 1. Introduction

Parkinson´s disease (PD) is the second most common neurodegenerative disease and the most frequent form of parkinsonism. The prevalence of PD in 2005 was about 4.1–4.6 million people over the age of 50 worldwide, and these numbers are expected to double to 8.7–9.3 million people by 2030 [1]. The diagnosis of PD is based on motor symptoms, i.e., rest tremor, rigidity, akinesia and postural instability [2]. However, a wide variety of non-motor symptoms also accompany the disease: neuropsychiatric, sleep, olfactory, autonomic, gastrointestinal and sensory symptoms [3]. As the disease progresses, other complications may appear such as psychosis, dementia, falls, trauma, dysphagia or infections. These complications negatively influence the quality of life and the burden of patients and caregivers, and result in higher costs and the utilization of healthcare resources [4]. 

Patients with PD are more prone to hospital admissions when compared with sex and age-matched controls [5], and have longer lengths of in-hospital stays (up to 2–14 days more) and increased complications and mortality [6,7]. The prevalence of comorbidity in PD is high and most admissions are caused by comorbid diseases [8,9]. The main reasons for hospitalization in PD include falls and fractures, syncope, cardiac diseases, pneumonia cognitive impairment, psychiatric disorders and adverse drug events [10]. The overall risk of in-hospital admission for a PD patient is approximately 7–28% per year [7].

It is of utter importance to measure and analyse national trends in PD hospitalizations, in order to allocate health resources and develop strategies to prevent or better manage these hospitalizations. Unwarranted admissions, complications and financial expenditures could be reduced by identifying the rates and causes of these admissions. There is growing evidence about overall PD hospitalizations in several countries [6,11,12,13,14,15,16], including Spain [17]. During the period 1997–2012, PD as a comorbid diagnosis during hospital stay progressively increased in Spain, but PD-related hospitalizations remained stable. However, several years have passed since those studies and data on hospital trends over the last decade (2010–2019) are lacking.

The Spanish National Hospital Discharge Database (SNHDD) covers most of the hospital admissions in the national healthcare system and provides information about clinical diagnoses related to such hospitalizations. Using the SNHDD, the objectives of this research were (1) to describe the overall trends in the incidence of hospitalizations of PD patients in Spain from 2010 to 2019; (2) to compare these incidences according to sex and diagnostic position of PD (“primary” or main reason of hospitalization; and “secondary” or PD as a comorbid condition during the hospitalization); and (3) to describe in-hospital mortality (IHM) of PD according to demographic factors such as time trends, age, sex and diagnostic position of PD. 

## 2. Materials and Methods

### 2.1. Study Design

This is a retrospective observational study whose main objective was to analyze the trends in the incidence of hospitalizations and in-hospital mortality of PD in Spain during the last decade (2010–2019). In addition, these data were also stratified according to a principal or secondary diagnosis of PD at admission, age, sex and overall comorbidity. 

### 2.2. Data Source

The study was performed using the Spanish National Hospital Discharge Database (SNHDD). This Spanish national database compiles all public hospital discharges, covering more than 95% of hospital admissions in Spain. The SNHDD applies the International Classification of Diseases, Ninth Revision, Clinical Modification (ICD-9-CM) diagnosis codes from 2010 to 2015; and the International Classification of Diseases, Tenth Revision (ICD-10), codes from 2016 to 2019. The database includes demographic variables such as sex and date of birth, epidemiologic variables such as admission and discharge dates, up to 14 (ICD-9-CM) or 20 (ICD-10) diagnoses at discharge and a list of the maximum 20 procedures performed during the hospital admission. All the specific details of the design and characteristics of the SNHDD are available elsewhere [18]. 

### 2.3. Study Population

Patients admitted to hospital with an age ≥18 years with a primary or secondary diagnosis of PD in the SNHDD were selected. In this regard, the diagnosis of PD was identified by the code 332.0 Parkinson´s disease (ICD-9-CM code) and G20 Parkinson´s disease (ICD-10 code). All data were collected from 1 January 2010 to 31 December 2019.

### 2.4. Study Variables

PD diagnosis was classified into a primary or a secondary diagnosis in relation to hospital admission. PD was classified as the primary diagnosis if the code appeared as the first diagnosis in the SNHDD database, and as a secondary diagnosis in case it appeared at any other diagnostic position (positions 2 to 14/20). Patients were also stratified by sex (male/female) and six age ranges (less than 40 years, 40–49, 50–59, 60–69, 70–79, more than 80 years). Data about comorbidities were retrieved from the diagnosis list of each patient with PD, using the method described by Quan et al. with the enhanced ICD-9-CM and ICD-10 [19]. Comorbidities were summarized for an easier interpretation by measuring the Charlson comorbidity index [20]. In-hospital mortality (IHM), defined as the proportion of patients who died during the admission, and the median of length of hospital stay (LOHS), were also estimated.

### 2.5. Statistical Analysis

Five biannual periods for the last decade were considered (2010–2011; 2012–2013; 2014–2015; 2016–2017; 2018–2019). The incidence rates of admission for PD were calculated per 100,000 inhabitants, dividing the number of cases per year, sex, and age group by the corresponding number of people in that population group, according to the data from the Spanish National Institute of Statistics (INS) reported on 31 December of each year [21]. Trends in hospitalizations were assessed by Poisson regression models adjusted by sex and age, if appropriate.

A log linear joinpoint regression was used to identify periods of change in annual hospitalization trends in PD, divided by sex and primary/secondary diagnosis. Joinpoint regression is a statistical analysis of time trends using joinpoint models, that is, models that fit the data to a trend, selecting the simplest model that the data allows, where several linear models are connected to each other in the points of intersection or joinpoint, forming linear splines [22]. The annual percentage of change (APC) was calculated in every period delimited by the points of change. A minimum number of joinpoints were initially performed; afterwards, the addition of consecutive joinpoints were added to test if they were statistically significant [22]. In the final model, each joinpoint indicated a significant trend change, and the APC was obtained in each of the segments delimited by the joinpoints, using the weighted least squares technique. The Joinpoint Regression Program, version 4.0.4 (National Cancer Institute, Bethesda, MD, USA), was used for the analysis [23].

A descriptive analysis was carried out for all categories and continuous variables. Variables were expressed as proportions, means with standard deviations and medians with interquartile ranges (e.g., LOHS), as appropriate. A bivariable analysis stratified by biannual periods was performed using the χ^2^ test for linear trend (proportions), ANOVA (means), and Kruskall–Wallis (medians), as appropriate. 

To identify an association between IHM and other variables measured as a binary outcome, three multivariable logistic regression analyses were performed, one for each diagnosis position of PD (primary, secondary, both). The variables included in the regression analyses were those with significant results in the bivariable analysis and/or those considered relevant. Estimates were presented as Odds Ratio (OR) with their 95% CI.

All statistical analysis was performed with Stata version 10.1 (Stata, College Station, TX, USA). Statistical significance was set at *p* < 0.05 (2-tailed).

### 2.6. Ethical Aspects

The SNHDD always maintains data confidentiality. Due to the anonymous nature of this database, it was not necessary to obtain informed consent or approval by an ethics committee in accordance with Spanish legislation. This study also followed the STROBE standard for reporting observational studies [24]. 

## 3. Results

The total number of hospitalizations with a diagnosis of PD in Spain were approximately 410,000 between 1 January 2010 and 31 December 2019. The proportion of female patients was 46.88% and the mean age was 79.62 (SD = 8.77). The vast majority of patients had PD as a secondary diagnosis (95.84%), while in only 4.16% of patients was PD the main reason for the hospitalization.

### 3.1. Overall Trends in PD Hospitalizations

As referenced in Table 1, a consistent and progressive increase in the number of PD coding in Spanish hospitalizations was observed during the last decade, ranging from 76,237 in 2010–2011 to 88,464 in 2018–2019 (i.e., from 81.25 cases to 94.82 cases per 100,000 inhabitants, respectively) (*p* < 0.001). Female sex decreased progressively over time (47.71% vs. 45.88%; *p* < 0.001). There was a slight but statistically significant increase in age (79.14 years in 2010–2011 vs. 80.05 in 2018–2019; *p* < 0.001). Remarkably, 57.18% of hospitalized PD patients had more than 80 years of age and the proportion of elderly patients increased over time (53.69% in 2010–2011 vs. 59.44% in 2018–2019). 

There was also a significant increase in overall comorbidity measured with the CCI index over the study period (mean CCI 1.55 in 2010–2011 vs. 1.79 in 2018–2019, *p* < 0.001). Accordingly, the proportion of PD patients with less comorbidities decreased (29.79% in 2010–2011 vs. 27.03% in 2018–2019) and PD patients with a high number of comorbidities increased (41.7% in 2010–2011 vs. 46.66% in 2018–2019) (*p* < 0.001) (Table 1). Median LOHS for hospitalizations for PD decreased from a mean of 7 days in 2010–2011 to 6 days, *p* < 0.001. 

### 3.2. Trends and Sex Differences in PD as Primary Diagnosis

As can be seen in Table 2 and Table 3, there has been an increase in PD coding as primary diagnosis in Spanish hospitalizations. This increase is represented in both sexes with a predominance in males (male population 3.81 vs. 4.41 per 100,000 inhabitants; female population 2.98 vs. 3.25 per 100,000 inhabitants). The results of the joinpoint analysis (Figure 1) showed that the admissions of patients with PD in a primary diagnosis position increased by 1.26% per year in both sexes, but this rise was only statistically significant in men (1.58% per year, *p* < 0.05).

There was a statistically significant increase in the mean age of male patients (68.84 years in the first period vs. 68.99 years in the last period, *p* = 0.005), while the mean age in females remained stable (*p* = 0.307). Thus, the mean age of patients hospitalized for PD as a primary diagnosis has remained unchanged in the last decade. In parallel, comorbidity according to the CCI has also remained unchanged for hospitalizations for a primary diagnosis of PD, both for males (*p* = 0.395) and females (*p* = 0.480). Indeed, the proportion of males hospitalized for PD with at least three comorbidities decreased statistically over time (*p* = 0.031, Table 2). 

Surprisingly, the in-hospital mortality was low, but increased slightly in both sexes during the study period (males: 2.34% in 2010–2011 vs. 3.59% in 2018–2019, *p* = 0.031; females: 2.17% in 2010–2011 vs. 3.2% in 2018–2019, *p* = 0.003). The length of hospital stay (LOHS) decreased slightly during the study period in both sexes (6 days in 2010–2011 vs. 5 days in 2018–2019; *p* <0.01).

### 3.3. Trends and Sex Differences in PD as Secondary Diagnosis

Table 4 and Table 5 show the data of hospitalizations of PD as a secondary diagnosis in men and women, respectively. There has been a remarkable increase in the diagnoses of PD as a comorbidity in hospitalizations during this study period for both sexes. This rise is more evident in men than women (male population 82.86 vs. 99.49 per 100,000 inhabitants; female population 73.06 vs. 82.7 per 100,000 inhabitants, *p* < 0.001). According to age groups, this increment was observed in men over 50 years and women over 60 years, but not in younger patients. The joinpoint analysis showed there has been a statistically significant increase in hospitalizations in both sexes, but again the increase was higher in men (APC: female 1.29% per year; men 2.07% per year). 

The mean age has also slightly increased in both sexes, being older in women than in men (males: 78.85 years in the first study period vs. 79.65 years the last study period, *p* < 0.001; females: 80.34 years vs. 81.45 years, *p* < 0.001). Comorbidity is also higher at the end of the period for both sexes (CCI in males: 1.74 in 2010–2011 vs. 1.98 in 2018–2019, *p* < 0.001; CCI females: 1.43 vs. 1.67, *p* < 0.001). Like in the overall trends, there has also been a slight increase in mortality both in males (11.42% vs. 11.71%, *p* = 0.02) and females (10.03% vs. 10.56%, *p* = 0.01). 

### 3.4. Factors Associated with In-Hospital Mortality

The overall IHM for PD patients in Spain in the last decade was 10.48% (42,897 in-hospital deaths), showing a slight but progressive statistically significant increase over time (10.4% in 2010–2011 vs. 10.87% in 2018–2019, *p* < 0.001; Table 1). Table 6 and Table 7 show the multivariate analysis for IHM regarding diagnostic position and sex, respectively. Independent factors associated to IHM were higher age, comorbidity (CCI), male sex and secondary diagnostic category. However, adjusted for possible confounders in the multivariate analysis, there was no significant change in IHM rates over the study period, except for an increase in mortality for men with a primary diagnosis of PD during the 2018–2019 period. Age had the most remarkable influence on mortality in both diagnostic categories and sexes, and this effect was higher in men and in PD as a secondary diagnosis of hospitalization.

## 4. Discussion

This epidemiological retrospective research analysed hospital discharges of PD patients during 2010–2019 in Spain. There was a constant and progressive increase in PD hospitalizations during the last decade, with a parallel rise of mean age, male sex and overall comorbidity. PD increased as a primary and a secondary diagnosis, but there was a higher increase of PD as a comorbid condition in relation to the main reason for hospitalization.

There is an ongoing interest in identifying the rates and causes of hospitalization, in order to reduce admissions, morbimortality and the financial impact associated with PD [25]. However, few studies have specifically analysed epidemiological hospitalization trends in the last decades [11,13,26]. An Irish report from 2009 to 2012 showed a steady increase in PD hospitalizations, especially over the age of 65 [26]. Tönges et al. also found an increase in parkinsonian syndromes in Germany during the period of 2010–2015, especially in rural areas [27]. In contrast, Mahajan et al. used the Nationwide Inpatient Sample (NIS) database in the United States to describe hospitalization trends from 2002 to 2011 [9]. In their study, a progressive reduction in the incidence of hospitalization was found during the whole study period, probably related to better outpatient and home care. Chana-Cuevas et al. have also described a steady decrease in PD hospitalizations in Chile from 2001 to 2018, in relation to the development of new government policies that guarantee an optimal outpatient management [6]. However, there is a lack of standardized epidemiological information about PD hospitalizations among countries around the world that enables any comparison among healthcare and economic systems.

One of the studies that shares a similar methodology with ours has been made in Spain in the period of 1997–2012 [15]. Gil-Prieto et al. showed there was a progressive increase in PD as a comorbidity, but PD-related hospitalizations remained stable. Our data confirms an ongoing ascending trend in PD hospitalizations during the last decade, not only in secondary but also in primary diagnoses. Gil-Prieto et al. also stated an increase in age and male sex, which is consistent with our findings. Thus, taking both study periods, the hospitalization rate for comorbid PD in Spain has more than doubled, going from 34.91 (1997) to 94.82 (2019) per 100,000 inhabitants. In contrast, the median LOHS has decreased from 14 days (1997) to 6 days (2019). The reasons for this might be an optimization of in-patient treatments, better organization of hospital stays and improvement in outpatient resources, such as the development of Movement Disorders units [28]. There is an overall fall in incidence of PD hospitalizations that is visible in all tables and in the joinpoint regression models from 2015 to 2016. We believe this decrease may be a product of the coding process, as in 2016 the Spanish Ministry of Health changed the coding system from ICD-9-CM to ICD-10 [18].

The increase in PD hospitalizations in our study may be in accordance with epidemiological trends in the overall prevalence of PD. Already in 2005, an epidemiological study based on the projected number of people with PD in several countries stated that the prevalence of PD will double by 2030 (from 4.1–4.6 million to 8.7–9.3 million people worldwide). In that research, Spain was also specifically estimated to double its PD prevalence (from 0.26 to 0.4 million people by 2030) [1]. Since then, there has been growing evidence of an increase in PD prevalence and incidence worldwide, with specific regional and country differences [29]. Although our study is not specifically designed to measure population prevalence, the steady rise in hospitalizations may be an indirect measure of an increasing prevalence of PD in Spain. However, caution must be taken into account, as hospitalization rates may be influenced by other factors, such as government health policies and outpatient resources.

Although our data initially suggested a slight increase in mortality over time, the multivariate analysis showed that in-hospital case fatality remained stable in the last decade after adjusting for confounding factors. Age, male sex and comorbidity resulted in independent factors of mortality that increased during the study period, which justifies the slight increase in overall mortality. In accordance with our data, other studies from the United States or Estonia have also found a stabilization in PD mortality [11,30]. In contrast, several studies have reported an increase in PD fatality in the last 20 years [31,32,33]. However, these latter studies did not consider overall comorbidity associated with the increasing prevalence of age, which might be an important confounding factor for crude mortality. Remarkably, there was a slight increase in adjusted mortality for men with a primary diagnosis of PD during the 2018–2019 period. We do not know the reason for this finding in this precise period and we must be cautious, as the sample size for hospitalizations regarding primary diagnoses of PD is relatively small. Future studies will confirm if this finding becomes a trend in the following decade.

Interestingly, PD as a secondary diagnosis was also an independent factor for mortality. This may imply that PD as a disease may worsen overall prognosis by itself, but also that the outcome of hospital care for people with PD may depend on the clinical team´s experience with PD, as secondary PD diagnosis might be predominantly coded by non-neurological departments [34]. This possibly underlines the importance of specialized PD hospital care and the need for PD-related education for non-PD-specialized healthcare professionals.

Regarding the influence of sex among PD hospitalizations, there has been an increase in male patients. This feature is in concordance with the epidemiological increase of the male sex among the overall PD prevalence [29]. Male sex has also become an independent factor for mortality, when adjusted for age and comorbidity. Thus, a higher mortality due to the increased prevalence of vascular risk factors and diseases among men does not explain the whole picture. Some studies have suggested that estrogen may increase physiological striatal dopamine levels, which may explain the lower incidence, higher age and more benign phenotype in women with PD [35].

The strength of our research lies in the extensive national representation of the database, with a uniform collection of data through ICD-9CM and ICD-10 codes over the last decade. Nevertheless, our study is not without limitations. First, the SNHDD has an administrative purpose and lacks the individual patient details needed to make accurate clinical associations. Second, our analysis has a retrospective design, which might be subject to selection bias and be enhanced by the moderate sensitivity of diagnostic codes for PD. A recent study performed in the UK found that PD was underestimated by 27% by national databases that used ICD coding [36]. Third, our data reflects the diagnosis of PD after in-hospital discharges but does not capture emergency department visits without in-hospital admissions. Finally, our design has a strict epidemiological purpose to describe national trends but lacks an analysis of other clinical variables and comparisons with a control group.

## 5. Conclusions

A steady rise in PD hospitalizations has been confirmed during the last decade in Spain (2010–2019), which maintains the previous national trend since 1997. The increase of PD has mainly occurred as a comorbid diagnosis, but also PD-related hospitalizations have risen. Age, male sex and comorbidity have progressively increased in PD patients. However, the adjusted mortality rate has been stable during the whole period and there has been a constant decrease in the lengths of hospital stays since 1997. It is important to analyze epidemiological trends in PD, as the burden of this disease in hospitalization may increase the complexity and costs of hospitalized patients in the future.

## Figures and Tables

**Figure 1 jcm-12-00902-f001:**
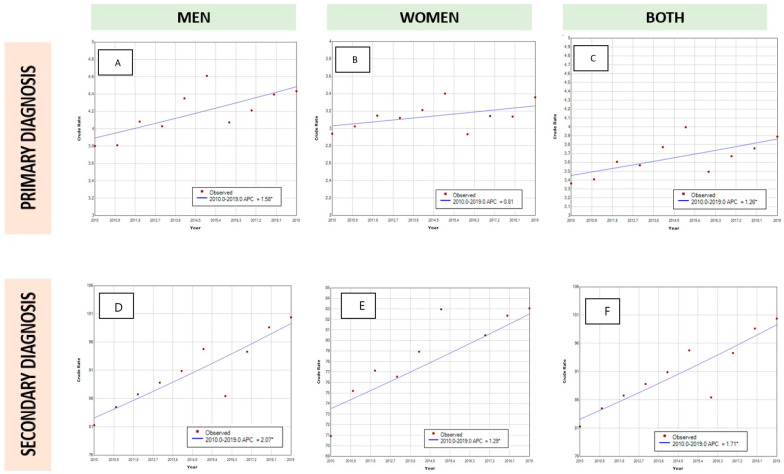
Joinpoint analysis with Annual Percent Change (APC) in PD as a primary diagnosis in men (**A**), women (**B**) and both sexes (**C**). Annual Percent Change (APC) in PD as a secondary diagnosis in men (**D**), women (**E**) and both sexes (**F**). * Indicates that the APC is significantly different from zero at the alpha = 0.05 level.

**Table 1 jcm-12-00902-t001:** Characteristics of Hospital Admissions of Parkinson´s Disease as a Primary or Secondary Diagnosis in Spain (2010–2019).

Variable		2010–2011	2012–2013	2014–2015	2016–2017	2018–2019	Total	*p*-Value
***n* (number of hospital admissions)**		76,237	79,772	84,194	80,811	88,464	409,478	<0.01 *
**Incidence per 100,000 population**		81.25	85.79	90.66	86.56	94.82	87.81	<0.01 *
***n* (%)**	**Primary diagnosis**	3177 (4.17)	3335 (4.18)	3606 (4.28)	3344 (4.14)	3567 (4.03)	17,029 (4.16)	0.135
	**Secondary diagnosis**	73,060 (95.83)	76,437 (95.82)	80,588 (95.72)	77,467 (95.86)	84,897 (95.97)	392,449 (95.84)	0.135
**Female sex, *n* (%)**		36,369 (47.71)	37,870 (47.47)	39,789 (47.26)	37,353 (46.22)	40,583 (45.88)	191,964 (46.88)	<0.01 *
**Age, years, Mean (SD)**		79.14 (8.54)	79.32 (8.66)	79.59 (8.77)	79.92 (8.85)	80.05 (8.98)	79.62 (8.77)	<0.01 *
**Age, subgroups *n* (%)**	**≤40 years**	134 (0.18)	113 (0.14)	108 (0.13)	105 (0.13)	115 (0.13)	575 (0.14)	0.018 *
	**40–49 years**	465 (0.61)	492 (0.62)	520 (0.62)	432 (0.53)	482 (0.54)	2391 (0.58)	0.003 *
	**50–59 years**	1537 (2.02)	1684 (2.11)	1810 (2.15)	1794 (2.22)	2043 (2.31)	8868 (2.17)	<0.01 *
	**60–69 years**	6572 (8.62)	7369 (9.24)	7585 (9.01)	6884 (8.52)	7470 (8.44)	35,880 (8.76)	<0.01 *
	**70–79 years**	26,596 (34.89)	25,851 (32.41)	25,791 (30.63)	23,639 (29.25)	25,775 (29.14)	127,652 (31.17)	<0.01 *
	**≥80 years**	40,933 (53.69)	44,263 (55.49)	48,380 (57.47)	47,957 (59.35)	52,579 (59.43)	234,112 (57.18)	<0.01 *
**CCI, Mean (SD)**		1.55 (1.6)	1.61 (1.62)	1.63 (1.64)	1.74 (1.73)	1.79 (1.78)	1.67 (1.68)	<0.01 *
**CCI, subgroups, *n* (%)**	**1**	22,711 (29.79)	22,863 (28.66)	24,039 (28.55)	22,052 (27.29)	23,913 (27.03)	115,578 (28.23)	<0.01 *
	**2**	21,733 (28.51)	22,467 (28.16)	23,311 (27.69)	21,933 (27.14)	23,275 (26.31)	112,719 (27.53)	
	**≥3**	31,793 (41.7)	34,442 (43.18)	36,844 (43.76)	36,826 (45.57)	41,276 (46.66)	181,181 (44.25)	
**LOHS, Median (IQR)**		7 (8)	7 (7)	6 (8)	6 (8)	6 (7)	6 (8)	<0.01 *
**IHM, *n* (%)**		7930 (10.4)	8114 (10.17)	8611 (10.23)	8626 (10.67)	9616 (10.87)	42,897 (10.48)	<0.01 *

CCI Charlson Comorbidity Index. LOHS Length of hospital stay. IHM In hospital mortality. * Statistically significant.

**Table 2 jcm-12-00902-t002:** Characteristics of Hospital Admissions of Male Patients with Parkinson´s Disease as a Primary Diagnosis in Spain (2010–2019).

Variable		2010–2011	2012–2013	2014–2015	2016–2017	2018–2019	Total	*p*-Value
**Male sex, *n***		1751	1851	2045	1905	2034	9586	<0.01 *
**Incidence per 100,000 population**		3.81	4.06	4.48	4.14	4.41	4.18	<0.01 *
**Age, years, Mean (SD)**		68.84 (12.53)	69.31 (11.74)	68.1 (11.65)	68.36 (11.74)	68.99 (12.31)	68.71 (12)	<0.01 *
**Age, subgroups *n* (%)**	**≤40 years**	30 (1.71)	22 (1.19)	22 (1.08)	29 (1.52)	25 (1.23)	128 (1.34)	0.555
	**40–49 years**	127 (7.25)	93 (5.02)	111 (5.43)	100 (5.25)	122 (6)	553 (5.77)	0.026 *
	**50–59 years**	224 (12.79)	259 (13.99)	328 (16.04)	292 (15.33)	304 (14.95)	1407 (14.68)	0.178
	**60–69 years**	440 (25.13)	510 (27.55)	587 (28.7)	503 (26.4)	507 (24.93)	2547 (26.57)	0.053
	**70–79 years**	568 (32.44)	573 (30.96)	640 (31.3)	654 (34.33)	658 (32.35)	3093 (32.27)	0.176
	**≥80 years**	362 (20.68)	394 (21.28)	357 (17.46)	327 (17.17)	418 (20.55)	1858 (19.38)	0.023 *
**CCI, Mean (SD)**		0.71 (1.16)	0.76 (1.26)	0.73 (1.15)	0.66 (1.15)	0.74 (1.29)	0.72 (1.2)	0.395
**CCI, subgroups, *n* (%)**	**1**	1063 (60.71)	1112 (60.08)	1226 (59.95)	1209 (63.46)	1240 (60.96)	5850 (61.03)	0.502
	**2**	367 (20.96)	374 (20.21)	438 (21.42)	407 (21.36)	454 (22.32)	2040 (21.28)	0.893
	**≥3**	321 (18.33)	365 (19.72)	381 (18.63)	289 (15.17)	340 (16.72)	1696 (17.69)	0.031 *
**LOHS, Median (IQR)**		6 (7)	5 (6)	5 (7)	5 (6)	5 (5)	5 (6)	<0.01 *
**IHM, *n* (%)**		41 (2.34)	41 (2.22)	45 (2.2)	50 (2.62)	73 (3.59)	250 (2.61)	0.031 *

CCI Charlson Comorbidity Index. LOHS Length of hospital stay. IHM In hospital mortality. * Statistically significant.

**Table 3 jcm-12-00902-t003:** Characteristics of Hospital Admissions of Female Patients with Parkinson´s Disease as a Primary Diagnosis in Spain (2010–2019).

Variable		2010–2011	2012–2013	2014–2015	2016–2017	2018–2019	Total	*p*-Value
**Female sex, *n***		1426	1484	1561	1439	1533	7443	<0.01 *
**Incidence per 100,000 population**		2.98	3.13	3.31	3.04	3.25	3.14	<0.01 *
**Age, years, Mean (SD)**		70.28 (11.41)	69.9 (11.86)	70.31 (11.16)	71.02 (10.64)	70.8 (10.46)	70.46 (11.12)	0.307
**Age, subgroups *n* (%)**	**≤40 years**	16 (1.12)	18 (1.21)	11 (0.7)	5 (0.35)	4 (0.26)	54 (0.73)	<0.01 *
	**40–49 years**	60 (4.21)	74 (4.99)	64 (4.1)	43 (2.99)	44 (2.87)	285 (3.83)	0.093
	**50–59 years**	162 (11.36)	147 (9.91)	161 (10.31)	175 (12.16)	173 (11.29)	818 (10.99)	0.492
	**60–69 years**	366 (25.67)	461 (31.06)	463 (29.66)	369 (25.64)	417 (27.2)	2076 (27.89)	0.02 *
	**70–79 years**	525 (36.82)	456 (30.73)	529 (33.89)	504 (35.02)	584 (38.1)	2598 (34.91)	<0.01 *
	**≥80 years**	297 (20.83)	328 (22.1)	333 (21.33)	343 (23.84)	311 (20.29)	1612 (21.66)	0.210
**CCI, Mean (SD)**		0.54 (0.93)	0.55 (0.95)	0.6 (0.99)	0.61 (1.08)	0.58 (1.05)	0.58 (1)	0.480
**CCI, subgroups, *n* (%)**	**1**	936 (65.64)	984 (66.31)	979 (62.72)	924 (64.21)	1003 (65.43)	4826 (64.84)	0.558
	**2**	305 (21.39)	299 (20.15)	359 (23)	329 (22.86)	321 (20.94)	1613 (21.67)	0.261
	**≥3**	185 (12.97)	201 (13.54)	223 (14.29)	186 (12.93)	209 (13.63)	1004 (13.49)	0.847
**LOHS, Median (IQR)**		6 (7)	5 (6)	5 (6)	5 (7)	5 (6)	5 (6)	<0.01 *
**IHM, *n* (%)**		31 (2.17)	25 (1.68)	27 (1.73)	48 (3.34)	49 (3.2)	180 (2.42)	<0.01 *

CCI Charlson Comorbidity Index. LOHS Length of hospital stay. IHM In hospital mortality. * Statistically significant.

**Table 4 jcm-12-00902-t004:** Characteristics of Hospital Admissions of Male Patients with Parkinson´s Disease as a Secondary Diagnosis in Spain (2010–2019).

Variable		2010–2011	2012–2013	2014–2015	2016–2017	2018–2019	Total	*p*-Value
**Male sex, *n***		38,117	40,051	42,360	41,553	45,847	207,928	<0.01 *
**Incidence per 100,000 population**		82.86	87.76	92.81	90.35	99.49	90.66	<0.01 *
**Age, years, Mean (SD)**		78.85 (8.08)	78.98 (8.25)	79.3 (8.42)	79.65 (8.45)	79.65 (8.64)	79.3 (8.39)	<0.01 *
**Age, subgroups *n* (%)**	**≤40 years**	43 (0.11)	40 (0.1)	54 (0.13)	41 (0.1)	49 (0.11)	227 (0.11)	0.720
	**40–49 years**	154 (0.4)	193 (0.48)	205 (0.48)	173 (0.42)	199 (0.43)	924 (0.44)	0.167
	**50–59 years**	710 (1.86)	762 (1.9)	803 (1.9)	789 (1.9)	1010 (2.2)	4074 (1.96)	<0.01 *
	**60–69 years**	3394 (8.9)	3828 (9.56)	3917 (9.25)	3671 (8.83)	4051 (8.84)	18,861 (9.07)	<0.01 *
	**70–79 years**	14,195 (37.24)	13,886 (34.67)	13,917 (32.85)	12,845 (30.91)	14,331 (31.26)	69,174 (33.27)	<0.01 *
	**≥80 years**	19,621 (51.48)	21,342 (53.29)	23,464 (55.39)	24,034 (57.84)	26,207 (57.16)	114,668 (55.15)	<0.01 *
**CCI, Mean (SD)**		1.74 (1.69)	1.8 (1.69)	1.82 (1.72)	1.93 (1.82)	1.98 (1.86)	1.86 (1.76)	<0.01 *
**CCI, subgroups, *n* (%)**	**1**	9728 (25.52)	9767 (24.39)	10,354 (24.44)	9763 (23.5)	10,726 (23.4)	50,338 (24.21)	<0.01 *
	**2**	10,514 (27.58)	10,826 (27.03)	11,296 (26.67)	10,766 (25.91)	11,493 (25.07)	54,895 (26.4)	<0.01 *
	**≥3**	17,875 (46.9)	19,458 (48.58)	20,710 (48.89)	21,024 (50.6)	23,628 (51.54)	102,695 (49.39)	<0.01*
**LOHS, Median (IQR)**		7(8)	7 (8)	6 (8)	6 (8)	6 (8)	6 (8)	<0.01*
**IHM, *n* (%)**		4352(11.42)	4450 (11.11)	4729 (11.16)	4816 (11.59)	5369 (11.71)	23,716 (11.41)	0.021*

CCI Charlson Comorbidity Index. LOHS Length of hospital stay. IHM In hospital mortality. * Statistically significant.

**Table 5 jcm-12-00902-t005:** Characteristics of Hospital Admissions of Female Patients with Parkinson´s Disease as a Secondary Diagnosis in Spain (2010–2019).

Variable		2010–2011	2012–2013	2014–2015	2016–2017	2018–2019	Total	*p*-Value
**Female sex, *n***		34,943	36,386	38,228	35,914	39,050	184,521	<0.01 *
**Incidence per 100,000 population**		73.06	76.85	80.95	75.82	82.7	77.86	<0.01 *
**Age, years, Mean (SD)**		80.34 (8.05)	80.59 (8.14)	80.9 (8.14)	81.2 (8.35)	81.45 (8.42)	80.91 (8.23)	<0.01 *
**Age, subgroups *n* (%)**	**≤40 years**	45 (0.13)	33 (0.09)	21 (0.05)	30 (0.08)	37 (0.09)	166 (0.09)	0.077
	**40–49 years**	124 (0.35)	132 (0.36)	140 (0.37)	116 (0.32)	117 (0.3)	629 (0.34)	0.408
	**50–59 years**	441 (1.26)	516 (1.42)	518 (1.36)	538 (1.5)	556 (1.42)	2569 (1.39)	0.051
	**60–69 years**	2372 (6.79)	2570 (7.06)	2618 (6.85)	2341 (6.52)	2495 (6.39)	12,396 (6.72)	<0.01 *
	**70–79 years**	11,308 (32.36)	10,936 (30.06)	10,705 (28)	9636 (26.83)	10,202 (26.13)	52,787 (28.61)	<0.01 *
	**≥80 years**	20,653 (59.11)	22,199 (61.01)	24,226 (63.37)	23,253 (64.75)	25,643 (65.67)	115,974 (62.85)	<0.01 *
**CCI, Mean (SD)**		1.43 (1.49)	1.48 (1.52)	1.52 (1.56)	1.61 (1.62)	1.67 (1.69)	1.54 (1.58)	<0.01 *
**CCI, subgroups, *n* (%)**	**1**	10,984 (31.43)	11,000 (30.23)	11,480 (30.03)	10,156 (28.28)	10,944 (28.03)	54,564 (29.57)	<0.01 *
	**2**	10,547 (30.18)	10,968 (30.14)	11,218 (29.34)	10,431 (29.04)	11,007 (28.19)	54,171 (29.36)	<0.01 *
	**≥3**	13,412 (38.38)	14,418 (39.63)	15,530 (40.62)	15,327 (42.68)	17,099 (43.79)	75,786 (41.07)	<0.01 *
**LOHS, Median (IQR)**		7 (8)	7 (7)	7 (7)	7 (7)	6 (7)	7 (7)	<0.01 *
**IHM, *n* (%)**		3506 (10.03)	3598 (9.89)	3810 (9.97)	3712 (10.34)	4125 (10.56)	18,751 (10.16)	0.010

CCI Charlson Comorbidity Index. LOHS Length of hospital stay. IHM In hospital mortality. * Statistically significant.

**Table 6 jcm-12-00902-t006:** Multivariate analysis of in-hospital mortality among patients with Parkinson´s disease codified as a primary or secondary diagnostic position. Spain, 2010–2019.

		Primary Diagnosis		Secondary Diagnosis		Both	
		OR	95% CI	OR	95% CI	OR	95% CI
**Year**	**2010–2011**	**1**	**-**	**1**		**1**	
	**2012–2013**	0.85	0.6–1.2	0.96	0.93–0.99	0.96	0.93–0.99
	**2014–2015**	0.89	0.64–1.25	0.95	0.92–0.98	0.92	0.89–0.96
	**2016–2017**	1.34	0.98–1.84	0.97	0.94–1.01	0.94	0.93–0.97
	**2018–2019**	1.54 *	1.14–2.08	0.98	0.95–1.02	0.95	0.92–0.99
**Age Group**	**<60 years**	1	-	1		1	
	**60–79 years**	1.84 *	1.25–6.58	7.24 *	3.55–9.19	5.26 *	2.95–7.76
	**80 or over**	8.42 *	1.36–25.83	16.53 *	6.55–33.36	12.76 *	3.96–29.64
**CCI**	**0**	1	-	1		1	
	**1**	1.88 *	1.48–2.37	1.39 *	1.35–1.43	1.4 *	1.02–1.63
	**2 O MAS**	1.65 *	1.28–2.14	1.81 *	1.76–1.86	1.77 *	1.69–1.85
**Sex**	**Female**	1	-	1		1	
	**Male**	1.14	0.94–1.39	1.15 *	1.13–1.18	1.15 *	1.01–1.35
**Diagnostic position**	**Primary**	not applicable		not applicable		1	
	**Secondary**	not applicable		not applicable		2.22 *	1.98–2.44

CCI Charlson Comorbidity Index. * Statistically significant.

**Table 7 jcm-12-00902-t007:** Multivariate analysis of in-hospital mortality among patients with Parkinson´s disease codified as a primary or secondary diagnostic position and stratified by sex. Spain, 2010–2019.

		MEN						WOMEN					
		Primary		Secondary		Both		Primary		Secondary		Both	
		OR	95% CI	OR	95% CI	OR	95% CI	OR	95% CI	OR	95% CI	OR	95% CI
**Year**	**2010–2011**	1	-	1		1		1		1		1	
	**2012–2013**	0.92	0.59–1.44	0.95	0.91–1	0.95	0.91–1	0.76	0.44–1.3	0.96	0.92–1.01	0.92	0.88–0.99
	**2014–2015**	1	0.65–1.54	0.95	0.91–0.99	0.95	0.91–0.99	0.75	0.44–1.27	0.95	0.91–1.00	0.91	0.85–0.97
	**2016–2017**	1.22	0.8–1.87	0.97	0.93–1.01	0.97	0.93–1.01	1.48	0.93–2.35	0.97	0.93–1.02	0.98	0.92–1.03
	**2018–2019**	1.57 *	1.06–2.33	0.98	0.94–1.02	0.99	0.95–1.03	1.51	0.95–2.4	0.99	0.94–1.03	0.95	0.90–1.01
**Age group**	**<60 years**	1	-	1	-	1	-	1	-	1	-	1	-
	**60–79 years**	1.62	0.85–12.35	14.96 *	2.38–42.87	9.85 *	2.45–39.61	2.02	0.87–4.71	1.99*	1.61–2.46	1.68 *	1.35–2.52
	**80 or over**	8.01 *	1.1–38.19	29.8 *	4.18–82.43	17.45 *	4.34–70.14	8.94 *	3.84–20.77	3.94*	3.2–4.85	3.36 *	2.6–5.04
**CCI**	**CCI = 0**	1	-	1	-	1	-	1	-	1	-	1	-
	**CCI = 1**	1.53 *	1.12–2.08	1.3 *	1.25–1.36	1.31 *	1.26–1.37	2.46 *	1.71–3.54	1.48*	1.42–1.55	1.43 *	1.42–1.52
	**CCI = 2 OR MORE**	1.34	0.96–1.85	1.66 *	1.6–1.72	1.66 *	1.6–1.72	2.24 *	1.49–3.36	2.01*	1.92–2.08	1.93 *	1.85–2.01
**Diagnostic position**	**Primary dx**	Not applicable		Not applicable		1		Not applicable		Not applicable		1	
	**Secondary dx**	Not applicable		Not applicable		2.98 *	2.62–3.39	Not applicable		Not applicable		2.65 *	2.28–3.08

CCI Charlson Comorbidity Index. * Statistically significant.

## Data Availability

The study was performed using the Spanish National Hospital Discharge Database (SNHDD). This Spanish national database compiles all public hospital discharges, covering more than 95% of hospital admissions in Spain. The SNHDD applies the International Classification of Diseases, Ninth Revision, Clinical Modification (ICD-9-CM) diagnosis codes from 2010 to 2015; and the International Classification of Diseases, Tenth Revision (ICD-10), codes from 2016 to 2019. The data from this registry is available and can be requested to the Ministry of Health upon reasonable request.

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
