# Peer review of "Trends and Sex Differences in Hospitalizations and Mortality in Parkinson’s Disease in Spain (2010–2019): A Nationwide Population-Based Study"

_jcm, 2023, doi:10.3390/jcm12030902_

Round 1
Reviewer 1 Report
This well-written and highly structured article presents the results of an epidemiological observational study that used nationwide administrative hospital discharge data to describe the trends in hospitalizations and in-hospital mortality of people with PD between 2010 and 2019 in Spain. The authors conclude that PD hospitalizations have increased with a parallel rise in mean age, and frequencies of male sex and comorbidities (not mentioned: secondary diagnoses) while in-hospital mortality adjusted for these factors remained unchanged. These results, which are based on scientifically sound methodology, are novel as they update current statistics on PD hospital care as a basis for appropriate health care planning. As the conclusions are mostly but only in part supported by the results and some minor details regarding reporting guidelines and presentation of results need improvement, a revision seems warranted.
MAJOR ISSUES
Discussion: when you write that “Age, male sex and comorbidity resulted in independent factors of mortality …” you do not mention secondary diagnosis type as an independent risk factor of in-hospital mortality. According to your data, being admitted with but not for PD seems to increase the risk of dying in the hospital. This may be a hint that the outcome of hospital care for people with PD depends on the clinical team’s awareness of and experience with PD as secondary PD diagnoses might be predominantly coded by non-neurological departments (ideally find and cite literature). This possibly underlines the importance of specialized PD hospital care and the need for PD-related education of non-PD-specialized healthcare professionals. You may consider emphasizing this finding and discussing it if deemed appropriate. At least, secondary diagnosis type as an independent risk factor of in-hospital mortality should be mentioned more clearly.
Discussion: you write that “The analysis showed that in-hospital case fatality remained stable in the last decade after adjusting by confounding factors.” Maybe, I am mistaken here, but doesn't the OR for in-hospital mortality of people with PD admitted to the hospital for PD (primary diagnosis) of the admission period 2018-2019 indicate an increase of adjusted in-hospital mortality in men with PD? (Table 7, first column, OR 1.57 (1.06-2.33)) Thus, there seems to be an increase in age- and comorbidity-adjusted in-hospital mortality of men between 2016/17 and 2018/19 which is not discussed or covered by the conclusions. If you agree that this is a relevant finding, you may consider discussing it.
MINOR ISSUES
Title Page: For the title to more properly reflect the subject of the article, you may consider adding “Trends and” to “Sex differences in hospitalizations and mortality …”
Abstract: Given the common reading habits of only skipping the abstract for important information, I feel that “this decade” could be described more precisely, e.g. “However, data on the trends during the following decade (2010-2019) are lacking.” For the same reason, you may consider adding “in PD inpatients” to “Male sex, age and comorbidity also increased progressively.”
Abstract: The important information “Adjusted mortality rate remained stable.” would benefit in precision if you added the factors the mortality rate was adjusted for in your final model to the abstract.
Abstract: You conclude by saying that “The burden of this disease may increase the complexity and costs of hospitalized patients in the future.” Complexity and costs are characteristics of PD hospital care rather than hospitalized PD patients. I would consider changing the wording.
Introduction: The phrase “The prevalence of comorbidity in PD is high…” could be strengthened in validity by adding references, e.g. https://doi.org/10.1186/s12883-017-0904-4 and https://doi.org/10.1016/j.parkreldis.2022.10.012.
Methods: Maybe, you could provide a short introductory sentence regarding joinpoint regression analyses for less experienced readers, characterizing them as a standard statistical approach to analyse trends in incidence and mortality.
Methods: For a reporting of your observational study aligned with best practice, you may consider applying the STROBE standard (von Elm et al.: Strengthening the Reporting of Observational Studies in Epidemiology. Lancet 2007 Oct 20;370(9596):1453-7. PMID: 18064739) and checking if all required information is displayed in the article.
Results: In general, there are many tables and figures that all significantly contribute to the overall argument. However, their readability could be improved. Maybe, you could add a simplified graph showing the temporal trends in PD hospitalization incidence per 100,000 population and in-hospital mortality throughout the study period and across the strata defined by the tables (i.e., primary or secondary and male or female, primary male, primary female, secondary male, secondary female).
Tables 1-5: readability can be improved by adding separate columns indicating the proportions which are the relevant parameter. Additionally, no data related to ‘emergency room’ are shown in the table such that ER should be removed from the table legends.
Figure 1: For a quicker grasp of the core content of your article, you may add row and column headings, i.e. ‘male, female, both’ as columns and ‘primary, secondary diagnosis’ as rows.
Results: The joinpoint regression models fitted lines to the crude PD coding rates that imply their steady increase (Fig. 1). However, across sexes and primary/secondary diagnosis type, there is a remarkable drop in APCs in 2015, seen in all diagrams. Do you consider this relevant and have an explanation?
Results: “In parallel, comorbidity according to the CCI has also remained unchanged for hospitalizations for a primary diagnosis of PD, both for males (p= 0.395) and females (p = 0.480).” In contrast to this statement that refers to the CCI mean only, you may add the information that the proportion of males hospitalized for PD with at least 3 comorbidities decreased over time (p = 0,031; Table 2).
Results: Fix typing error in “According to age groups, this increment *in* observed in men over 50 years…” – assuming “was” is the correct word.
Tables 6-7: You should correctly indicate the reported analysis as multivariate instead of multivariant. In addition, you should label the columns ‘OR’ and ‘95% CI’ and ideally include separate columns for ORs and CIs, as proposed by the APA rules of style (https://apastyle.apa.org/style-grammar-guidelines/tables-figures/sample-tables#regression). Additionally, you should add asterisks to the significantly elevated odds of in-hospital mortality for people admitted with PD as a secondary diagnosis in Table 7 and remove the incorrect asterisk next to the OR/CI “1.22(0.8-1.87)*” for the factor ‘year 2016-2017’.
Discussion: “However, few studies have specifically analysed epidemiological hospitalization trends in the last decades.” Here, you may consider adding and discussing the following work from Germany: https://doi.org/10.1007/s00115-018-0590-5
References: The publication year of ref. 15 (Shahgholi et al.) should be corrected from 207 to 2017.
Author Response
Dear Reviewer 1
We appreciate the time and effort that has been put into our manuscript, and we are grateful for the valuable comments and improvements to our document. We have incorporated the suggestions made. Those changes have been highlighted in the manuscript. See below all point-by-point responses to your comments and concerns. All page numbers refer to the revised manuscript file with changes.
MAJOR ISSUES
Comment 1: Discussion: when you write that “Age, male sex and comorbidity resulted in independent factors of mortality …” you do not mention secondary diagnosis type as an independent risk factor of in-hospital mortality. According to your data, being admitted with but not for PD seems to increase the risk of dying in the hospital. This may be a hint that the outcome of hospital care for people with PD depends on the clinical team’s awareness of and experience with PD as secondary PD diagnoses might be predominantly coded by non-neurological departments (ideally find and cite literature). This possibly underlines the importance of specialized PD hospital care and the need for PD-related education of non-PD-specialized healthcare professionals. You may consider emphasizing this finding and discussing it if deemed appropriate. At least, secondary diagnosis type as an independent risk factor of in-hospital mortality should be mentioned more clearly.
We thank the reviewer for this comment. We totally agree it has not been clearly mentioned and it was valuable implications in our discussion. We have added the following test:
Page 7:
Interestingly, PD as a secondary diagnosis was also an independent factor for mortality. This may imply that PD as a disease may worsen overall prognosis by itself, but also that the outcome of hospital care for people with PD may depend on the clinical team´s experience with PD, as secondary PD diagnosis might be predominantly coded by non-neurological departments [31]. This possibly underlines the importance of specialized PD hospital care and the need for PD-related education for non-PD-specialized healthcare professionals.
[31] Willis AW, Schootman M, Tran R, Kung N, Evanoff BA, Perlmutter JS, Racette BA. Neurologist-associated reduction in PD-related hospitalizations and health care expenditures. Neurology 2012; 79:1774-80.
Comment 2: Discussion: you write that “The analysis showed that in-hospital case fatality remained stable in the last decade after adjusting by confounding factors.” Maybe, I am mistaken here, but doesn't the OR for in-hospital mortality of people with PD admitted to the hospital for PD (primary diagnosis) of the admission period 2018-2019 indicate an increase of adjusted in-hospital mortality in men with PD? (Table 7, first column, OR 1.57 (1.06-2.33)) Thus, there seems to be an increase in age- and comorbidity-adjusted in-hospital mortality of men between 2016/17 and 2018/19 which is not discussed or covered by the conclusions. If you agree that this is a relevant finding, you may consider discussing it.
Thank you for this comment. Indeed, there is a slight increase in mortality in men hospitalized for PD as the main diagnosis during the 2018-2019, compared to the 2010-2011 period. We must be cautious with these findings, as sample size for primary diagnosis hospitalizations is relatively small. However, we consider this finding needs a mention in our article.
Page 6:
Remarkably, the was a slight increase in adjusted mortality for men with a primary diagnosis of PD during the 2018-2019 period. We do not know the reason for this finding in this precise period and we must be cautious, as the sample size for hospitalizations regarding primary diagnosis of PD is relatively small. Future studies will confirm if this finding becomes a trend in the following decade.
MINOR ISSUES
Comment 3: Title Page: For the title to more properly reflect the subject of the article, you may consider adding “Trends and” to “Sex differences in hospitalizations and mortality …”
We thank the suggestion. We have changed the title.
Comment 4: Abstract: Given the common reading habits of only skipping the abstract for important information, I feel that “this decade” could be described more precisely, e.g. “However, data on the trends during the following decade (2010-2019) are lacking.” For the same reason, you may consider adding “in PD inpatients” to “Male sex, age and comorbidity also increased progressively.”
Comment 5: Abstract: The important information “Adjusted mortality rate remained stable.” would benefit in precision if you added the factors the mortality rate was adjusted for in your final model to the abstract.
Comment 6: Abstract: You conclude by saying that “The burden of this disease may increase the complexity and costs of hospitalized patients in the future.” Complexity and costs are characteristics of PD hospital care rather than hospitalized PD patients. I would consider changing the wording.
We thank for the suggestions to the abstract. We have made the changes.
Comment 7: Introduction: The phrase “The prevalence of comorbidity in PD is high…” could be strengthened in validity by adding references, e.g. https://doi.org/10.1186/s12883-017-0904-4 and https://doi.org/10.1016/j.parkreldis.2022.10.012.
We believe they are two valuable references; we have included them.
Comment 8: Methods: Maybe, you could provide a short introductory sentence regarding joinpoint regression analyses for less experienced readers, characterizing them as a standard statistical approach to analyse trends in incidence and mortality.
An explanatory sentence has been added to Page 3:
Jointpoint regression is a statistical analysis of time trends using joinpoint models, that is, models that fit the data to a trend, selecting the simplest model that the data allows, where several linear models are connected to each other in the points of intersection or joinpoint, forming linear splines [23]
Comment 9: Methods: For a reporting of your observational study aligned with best practice, you may consider applying the STROBE standard (von Elm et al.: Strengthening the Reporting of Observational Studies in Epidemiology. Lancet 2007 Oct 20;370(9596):1453-7. PMID: 18064739) and checking if all required information is displayed in the article.
The STROBE checklist was used and uploaded in the submission process. However, we have also mentioned this in the manuscript in Page 3.
Comment 10: Results: In general, there are many tables and figures that all significantly contribute to the overall argument. However, their readability could be improved. Maybe, you could add a simplified graph showing the temporal trends in PD hospitalization incidence per 100,000 population and in-hospital mortality throughout the study period and across the strata defined by the tables (i.e., primary or secondary and male or female, primary male, primary female, secondary male, secondary female).
Comment 11: Tables 1-5: readability can be improved by adding separate columns indicating the proportions which are the relevant parameter. Additionally, no data related to ‘emergency room’ are shown in the table such that ER should be removed from the table legends.
We appreciate this suggestion. Unfortunately, we have tried these changes before submitting the article, but the readability was difficult. The graphics do not reflect the steady increase, as there is much difference among the incidence of primary/secondary diagnosis and the scale goes awry. That would mean designing several graphics, which would add complexity to the presentation of the article. Also, adding more columns to tables would made the numbers even smaller. We have deleted “ER” from the table legends.
Comment 12: Figure 1: For a quicker grasp of the core content of your article, you may add row and column headings, i.e. ‘male, female, both’ as columns and ‘primary, secondary diagnosis’ as rows.
Thank you for the recommendation. We have made the modification to the Figure in Page 5.
Comment 13: Results: The joinpoint regression models fitted lines to the crude PD coding rates that imply their steady increase (Fig. 1). However, across sexes and primary/secondary diagnosis type, there is a remarkable drop in APCs in 2015, seen in all diagrams. Do you consider this relevant and have an explanation?
Yes indeed, there is an overall fall in PD hospitalizations in all sexes and diagnostic positions from 2015 to 2016, which also reflects in the jointpoint graphics. We have made a modification in the discussion arguing this finding.
Page 6:
There is an overall fall in incidence of PD hospitalizations that is visible in all tables and in the jointpoint regression models from 2015 to 2016. We believe this decrease may be an artifact in the coding process, as in 2016 the Spanish Ministry of Health changed the coding system from ICD-9-CM to ICD-10 [19].
Comment 14: Results: “In parallel, comorbidity according to the CCI has also remained unchanged for hospitalizations for a primary diagnosis of PD, both for males (p= 0.395) and females (p = 0.480).” In contrast to this statement that refers to the CCI mean only, you may add the information that the proportion of males hospitalized for PD with at least 3 comorbidities decreased over time (p = 0,031; Table 2).
We did this note in Page 4.
Comment 15: Results: Fix typing error in “According to age groups, this increment *in* observed in men over 50 years…” – assuming “was” is the correct word.
Corrected.
Comment 16: Tables 6-7: You should correctly indicate the reported analysis as multivariate instead of multivariant. In addition, you should label the columns ‘OR’ and ‘95% CI’ and ideally include separate columns for ORs and CIs, as proposed by the APA rules of style (https://apastyle.apa.org/style-grammar-guidelines/tables-figures/sample-tables#regression). Additionally, you should add asterisks to the significantly elevated odds of in-hospital mortality for people admitted with PD as a secondary diagnosis in Table 7 and remove the incorrect asterisk next to the OR/CI “1.22(0.8-1.87)*” for the factor ‘year 2016-2017’.
We have done all the changes.
Comment 17: Discussion: “However, few studies have specifically analysed epidemiological hospitalization trends in the last decades.” Here, you may consider adding and discussing the following work from Germany: https://doi.org/10.1007/s00115-018-0590-5
We have added this reference in Page 6.
Comment 18: References: The publication year of ref. 15 (Shahgholi et al.) should be corrected from 207 to 2017.
Corrected.
Reviewer 2 Report
I commend the authors for analyzing the Parkinson's disease and patience acoustics in Spain. And I want to thank the authors and the other for allowing me to review this paper. The paper is well written with good english, and the paper is easy to read. The paper has appropriate references.
However, I feel that the discussion could be expanded by discussing Parkinson's disease trends in other European countries, and other countries around the world and discussing how their economic, social and medical facilities compare to each other, and how they might influence the statistics for Parkinson's disease hospitalizations in each country. The authors touch on this a little bit, but I think this should be the meat of the discussion and that it should be expanded more. For this article to be useful to physicians outside of Spain, this needs to be emphasized. The authors also need to expand on in the discussion of possible explanations of their findings, e.g. why there is a discrepancy in sex in Parkinson's disease hospitalizations in Spain?
please fix this spelling mistake: has been confirmed during the las decade.
I commend the authors for writing a good paper. The above are just my suggestions that I think would further improve the paper.
Author Response
Dear Reviewer 2
We appreciate the time and effort that has been put into our manuscript, and we are grateful for the valuable comments and improvements to our document. We have commented on the suggestions made.
We have broaden the information about PD hospitalizations with new references from other countries, as ref [28]. However, information comparing hospitalizations among countries and making a relation to economic, healthcare and social factoros are lacking. We have made also a mention to this in the discussion that could encourage further research, which, as the reviewer suggest, is of utter importance.
We have also discussed the role of sex in PD hospitalizations and the role of male sex in mortality in PD.
We would like to thank the reviewer for his/her valuable comments that have improved the quality of our manuscript.